# The Weak Relationship between Vitamin D Compounds and Glucose Homeostasis Measures in Pregnant Women with Obesity: An Exploratory Sub-Analysis of the DALI Study

**DOI:** 10.3390/nu14163256

**Published:** 2022-08-09

**Authors:** Lilian Cristina Mendoza, Jürgen Harreiter, Gernot Desoye, David Simmons, Juan M. Adelantado, Alexandra Kautzky-Willer, Agnieszka Zawiejska, Ewa Wender-Ozegowska, Annunziata Lapolla, Maria G. Dalfra, Alessandra Bertolotto, Roland Devlieger, Fidelma Dunne, Elisabeth R. Mathiesen, Peter Damm, Lisse Lotte Andersen, Dorte Moller Jensen, David Hill, Mireille Nicoline Maria van Poppel, Rosa Corcoy

**Affiliations:** 1Institut de Recerca de l’Hospital de la Santa Creu i Sant Pau, 08025 Barcelona, Spain; 2CIBER de Bioingeniería, Biomateriales y Nanomedicina (CIBER-BBN), 28029 Madrid, Spain; 3Clinical Division of Endocrinology and Metabolism, Department of Internal Medicine III, Gender Medicine Unit, Medical University of Vienna, Waehringer Guertel 18–20, 1090 Vienna, Austria; 4Department of Obstetrics and Gynecology, Medical University of Graz, 8036 Graz, Austria; 5Macarthur Clinical School, School of Medicine, Western Sydney University, Campbelltown, NSE 2560, Australia; 6Department of Reproduction, Poznan University of Medical Sciences, 60-525 Poznan, Poland; 7Department of Medicine, University of Padova, 35128 Padova, Italy; 8Department of Clinical and Experimental Medicine, University of Pisa, 56126 Pisa, Italy; 9Obstetrics and Gynecology, University Hospitals KU Leuven, 3000 Leuven, Belgium; 10College of Medicine, Nursing and Health Sciences, School of Medicine, National University of Ireland, H91 TK33 Galway, Ireland; 11Center for Pregnant Women with Diabetes, Departments of Endocrinology and Obstetrics, Rigshospitalet, University of Copenhagen, DK-1165 Copenhagen, Denmark; 12Department of Gynecology and Obstetrics, Odense University Hospital, 5000 Odense, Denmark; 13Lawson Health Research Institute, St. Joseph Health Care, London, ON N6A 4V2, Canada; 14Institute of Sport Science, University of Graz, 8010 Graz, Austria; 15Departament de Medicina, Universitat Autònoma de Barcelona, Bellaterra, 08193 Barcelona, Spain

**Keywords:** vitamin D compounds, 25OHD2, 25OHD3, C3-epimer, glucose homeostasis, pregnancy, obesity

## Abstract

Studies on the relationship between vitamin D (VitD) and glucose homeostasis usually consider either total VitD or 25OHD3 but not 25OHD2 and epimers. We aimed to evaluate the cross-sectional association of VitD compounds with glucose homeostasis measurements in pregnant women with overweight/obesity participating in the Vitamin D And Lifestyle Intervention for Gestational Diabetes Mellitus Prevention study. Methods: The analysis included 912 women. Inclusion criteria: <20 weeks gestation, body mass index ≥29 kg/m^2^ and information on exposure and outcome variables at baseline. Measurements: A 75 g OGTT at <20, 24–28 and 35–37 weeks gestation (except if previous diabetes diagnosis). Exposure variables: 25OHD2, 25OHD3 and C3-epimer. Outcome variables: fasting and post-challenge insulin sensitivity and secretion indices, corresponding disposition indices (DI), plasma glucose at fasting and 1 and 2 h, hyperglycemia in pregnancy (HiP). Statistics: Multivariate regression analyses with adjustment. Results: Baseline VitD sufficiency was 66.3%. Overall, VitD compounds did not show strong associations with any glucose homeostasis measures. 25OHD3 showed direct significant associations with: FPG at <20 and 24–28 weeks (standardized β coefficient (β) 0.124, *p* = 0.030 and 0.111, *p* = 0.026 respectively), 2 h plasma glucose at 24–28 weeks (β 0.120, *p* = 0.018), and insulin sensitivity (1/HOMA-IR, β 0.127, *p* = 0.027) at 35–37 weeks; it showed an inverse association with fasting DI (QUCKI*HOMA-β) at <20 and 24–28 weeks (β −0.124, *p* = 0.045 and β −0.148, *p* = 0.004 respectively). 25OHD2 showed direct associations with post-challenge insulin sensitivity (Matsuda, β 0.149, *p* = 0.048) at 24–28 weeks) and post-challenge DI (Matsuda*Stumvoll phase 1) at 24–28 and 35–37 weeks (β 0.168, *p* = 0.030, β 0.239, *p* = 0.006). No significant association with C3-epimer was observed at any time period. Conclusions: In these women with average baseline VitD in sufficiency range, VitD compounds did not show clear beneficial associations with glucose homeostasis measures.

## 1. Introduction

Vitamin D (VitD), a cholesterol-derived hormone mainly involved in calcium homeostasis, has been shown to have pleotropic effects including glucose metabolism [1]. 

VitD is available in two forms, ergocalciferol (VitD2), mainly derived from mushrooms and fortified foods, and cholecalciferol (VitD3), obtained from different dietary sources, mainly oil-rich fish or by skin synthesis from 7-dehydrocholesterol under the influence of UV light. While both are commonly used in nutritional supplements, certain differences according to VitD compound have been described. VitD enters the circulation bound to the vitamin D binding protein and is then hydroxylated to 25-hydroxycholecalciferol (25OHD), the main circulating form, in the liver. 25OHD3 binds with greater affinity to VitD-binding protein and is more efficiently hydroxylated and converted to 1,25-dihydroxycholecalciferol (1,25OH_2_D) than 25OHD2 [2,3]. Thus, 25OHD3 seems to display greater biological activity compared with 25OHD2 and has been shown to be more efficient in raising total 25OHD concentrations [4]. Therefore, supplementation studies commonly use 25OHD3.

In recent years, there has been growing interest in exploring the role of VitD in glucose metabolism. In vitro, 1,25OH_2_D stimulates the expression of insulin receptors [5,6] and modulates cytokine expression and activity [7,8], hence improving insulin sensitivity. On the other hand, in animal models, VitD deficiency has been shown to impair glucose-mediated insulin secretion [9,10,11], which can be restored after 1,25OH_2_D3 supplementation [12]. In in vitro models, 1,25OH_2_D3 enhances glucose-stimulated insulin secretion (GSIS) via calcium channel up-regulation [13]. Changes in β-cell gene expression affecting viability and apoptosis that also enhance GSIS have also been reported in cell lines treated with 1,25OH_2_D2 [14].

3-epi-25OHD3 (C3-epimer) results from the epimerization of 25OHD3 and represents 3.5 to 7% of the total 25OHD3 concentration in the general adult population, reaching up to 26% in a pediatric population [15]. In pregnancy, a sub-analysis from the Hyperglycemia And Pregnancy Outcomes study reported that C3-epimer can account for ~20% of 25OHD3 concentrations [16]. Recent publications support a possible biological role for C3-epimer. It has been reported that C3-epimer can suppress parathyroid hormone secretion and modulate cell differentiation and apoptosis [17,18]. Inverse associations have been described between C3-epimer and body mass index (BMI) and low density lipoprotein cholesterol [19]. Regarding glucose homeostasis, in 2019, Zheng et al. observed in the EPIC-InterAct case cohort an inverse association between non-epimeric 25OHD3 and the incidence of type 2 diabetes mellitus, while the C3-epimer showed a direct association. No association was observed between 25OHD2 and incidence of type 2 diabetes mellitus [20]. In pregnancy, an abstract published in 2019 observed no association between either 23OHD3 or C3-epimer and the risk of gestational diabetes mellitus (GDM) [21].

Circulating total 25OHD concentrations have been inversely associated with fasting plasma glucose (FPG) and essentially with improvements in insulin resistance [22,23,24,25]. However, randomized controlled trials of VitD3 supplementation aiming at type 2 diabetes mellitus prevention have had inconsistent results [26,27,28,29]. Studies investigating the effect of VitD3 on glucose homeostasis usually address FPG and insulin sensitivity and less often include measures of β-cell function. Improvements in FPG [26], insulin resistance, and 2 h plasma glucose [26,27] have been described in some trials, while others have not observed such benefits [30,31]. Insulin sensitivity and secretion can be separately measured, but it is the paired secretion-sensitivity relationship that is relevant for glucose homeostasis. The disposition index reflects insulin response at a given insulin sensitivity, which makes it a useful integrated β-cell function measure. Regarding β-cell function, a study by Mitri in 2011 described an improvement in the disposition index that was dependent on improved insulin secretion but not insulin sensitivity [32]. However, more recent studies have observed no effect in these parameters [33,34]. 

In pregnancy, VitD supplementation trials for GDM prevention have been conducted using VitD3 or a non-specified VitD [35,36,37,38] showing *potential benefits* according to the latest Cochrane review [39]. A beneficial effect on FPG and fasting insulin sensitivity has consistently been reported [40,41]. There are scarcer data on insulin secretion, but a recent metanalysis found an improvement in HOMA-β and 2 h plasma glucose [41]. 

To assess glucose homeostasis, several oral glucose tolerance test (OGTT)-derived indices to measure insulin sensitivity and secretion have been used in pregnancy studies (HOMA-IR, QUICKI, Matsuda, OGIS, HOMA-β, IGI, AUC_ins/glu_, Stumvoll), but only some of them have been validated during pregnancy. Kirwan et al. described a correlation between clamp and OGTT-derived HOMA-IR, QUICKI, and Matsuda indices, with the Matsuda index showing the strongest correlation [42]. As for insulin secretion, a recent report by Powe et al. evaluated different indices and concluded that Stumvoll phase 1 and AUC_ins/glu_ were valid OGTT-based (vs. clamp) insulin secretory response measures for pregnancy studies [43]. HOMA-β, the only fasting insulin secretion index evaluated, showed a weak positive correlation with first-phase insulin response in early pregnancy but not in late pregnancy [43].

In this exploratory sub-analysis of the Vitamin D And Lifestyle Intervention for Gestational Diabetes Mellitus Prevention (DALI) study, we aimed to evaluate the association of the serum concentrations of different VitD compounds with glucose homeostasis measures (FPG and post-challenge plasma glucose and insulin sensitivity, insulin secretion and disposition indices).

## 2. Materials and Methods

### 2.1. Study Population

A total of 984 women were recruited to participate in the DALI trials [38,44,45]. The full study protocol has been previously published [46]. In brief, the DALI project was a multicenter study conducted in nine European countries testing different strategies for GDM prevention. The study was approved by the ethics committees of all participating sites (NRES Committee East of England-Norfolk: 11/EE/0221; Medical University Poznan: 1165/12; UZ KU Leuven: ML7625; Hospital De La Santa Creu i Sant Pau Barcelona 13/006 (OBS); Medical University Vienna: 2022/2012-1369/2013; Province of Padua and Pisa: 4201 Å~11; Galway University Hospitals: 7/12; Ethical comittee VU medisch centrum Amsterdam: nr. 2012/400; Copenhague and Odense: Scientific Ethics Committee for the Capital Region, Hillerod, Denmark, Protokol nr.: H-4-2013-005). Women with <20 weeks’ gestation and BMI ≥29 Kg/m^2^ who signed a written consent were eligible for the study. Exclusion criteria were being unable to walk at least 100 m safely or to speak the language of the recruitment site, having complex diet requirements and chronic medical or psychiatric conditions, and, for the VitD trial, having current or past abnormal calcium metabolism or having hypercalciuria or hypercalcemia detected at baseline measurement.

All participants underwent a 75-g, 2 h OGTT at <20 weeks using IADPSG/WHO2013 criteria for GDM diagnosis. Women with normal glucose tolerance were randomized to three lifestyle arms [44]: lifestyle vs usual care [45] or lifestyle and/or VitD vs. usual care [38]. The lifestyle intervention consisted of healthy eating (promoting a high-fiber diet, lower in simple and complex carbohydrate, lower in fat, and limited intake of total calories), physical activity (promoting both aerobic and resistance activity according to American College of Obstetricians and Gynecologists (ACOG) guidelines), and combined healthy eating and physical activity interventions. Participants who signed informed consent and were excluded from the original trial allowed their data to be used in secondary analyses.

In this exploratory observational sub-analysis, all women (*n* = 912) with available OGTT and VitD data at baseline were included.

### 2.2. Data Collection and Assessments

Data from participating women were collected at 3 time points: <20 weeks (baseline), at 24–28, and at 35–37 weeks’ gestation.

Sociodemographic data and medical history were recorded at baseline. Blood samples and anthropometric measures were obtained at each time point. A standard 75-g, 2 h OGTT, after 10 h fasting, was performed at each time point (unless women had been previously diagnosed with overt diabetes/GDM). GDM was defined after IADPS/WHO 2013 diagnostic criteria (FPG ≥ 5.1 mmol/L and/or 1 h plasma glucose ≥ 10 mmol/L and/or 2 h plasma glucose ≥ 8.5 mmol/L). Overt diabetes was diagnosed if FPG ≥ 7.0 mmol/L or 2 h plasma glucose ≥ 11.1 mmol/L. Hyperglycemia in pregnancy (HiP) was defined as either GDM or overt diabetes. Blood samples were analyzed at local and central laboratories, with local results being used for clinical management. For the current analysis, central laboratory values were used (local data when central values were unavailable).

### 2.3. Measurements

During the OGTT, blood samples were drawn for the measurement of glucose at fasting and at 1 and 2 h (and additionally at 30 and 90 min in some study sites).

Glucose was measured using the hexokinase method (DiaSys Diagnostic Systems, Holzheim, Germany) with a lower limit of sensitivity of 0.1 mmol/L. Insulin was measured by a sandwich immunoassay (ADVIA Centaur; Siemens Health Care Diagnostics Inc., Vienna, Austria) with an analytical sensitivity of 0.5 mU/L, intra-assay coefficient of variation of 3.3% to 4.6%, and inter-assay coefficient of variation of 2.6% to 5.9%.

The following pregnancy-validated glucose homeostasis indices were calculated:
Insulin sensitivity at fasting using Homeostasis model assessment (HOMA-IR) [47] and Quantitative insulin check index (QUICKI) [48] and post-challenge using Matsuda and De Fronzo’s (ISI (comp)) [49].Insulin secretion at fasting using Homeostasis model assessment β (HOMA-β) [47] and post-challenge using Stumvoll phase 1 [50] and area under the curve insulin/glucose (AUC_ins/glu_) [43].The corresponding disposition indices: at fasting, QUICKI* HOMA-β and 1/HOMA-IR* HOMA-β and post-challenge, Matsuda*AUC_ins/glu_, and Matsuda*Stumvoll phase 1.Additional commonly used glucose homeostasis indices were also calculated as complementary information.Post-challenge insulin sensitivity using oral glucose insulin sensitivity (OGIS) [51].Postchallenge insulin secretion using early insulinogenic index (IGI) [52] and Stumvoll phase 2 [50].The corresponding disposition index (OGIS*IGI) plus those for combining OGIS and IGI with pregnancy-validated indices (Matsuda*IGI, OGIS*AUC_ins/glu_, OGIS*Stumvoll phase 1).

Serum 25(OH)D concentrations were measured using a ClinMass^®^ liquid chromatography–mass spectrometry/mass spectrometry (LCeMS/MS) complete kit (RECIPE Chemicals. Instruments GmbH, Munich, Germany). Vitamin D compounds 25OHD2, 25OHD3 were quantified, and C3-epimerwas qualitatively measured. All measurements were performed in the central laboratory.

### 2.4. Statistical Analysis

Categorical variables are presented as counts and percentages and continuous variables as median and percentile 25–75. At visual inspection, distributions did not substantially deviate from normal. 

Multivariate regression analyses (forward method) were used to assess the relationships between individual VitD compounds (25OHD2, 25OHD3 and C3-epimer) and glucose homeostasis variables: FPG, 1 and 2 h plasma glucose, pregnancy-validated indices (HOMA-IR, QUICKI, Matsuda, HOMA-β, Stumvoll phase 1, AUC_Ins/Glu_), additional glucose homeostasis indices (OGIS, IGI, Stumvoll phase 2), and fasting and post-challenge disposition indices at each pregnancy period evaluated. At <20 weeks, the regression model included VitD compounds (25OHD2, 25OHD3 and C3-epimer), age, BMI (at the time of evaluation), ethnicity, family history of diabetes, prior GDM, and recruitment site. At 24–28 and 35–37 weeks, the regression model included the abovementioned variables and DALI lifestyle intervention. 

Statistical assumptions for regression analyses were checked including lack of influential case detection (Cook’s distance < 1), collinearity diagnosis (variable inflation scores < 10), and residual independence (Durbin–Watson statistic between 1 and 3). 

Logistic regression analysis with the same exposure variables was used to assess the relationships between VitD compounds and HiP. Significance was defined as a two-sided *p* < 0.05. As this was an exploratory analysis, no sample size calculation was performed, and corrections for multiple comparisons were not applied.

To explore if vitamin D compounds had different impacts on glucose homeostasis variables in participants with different vitamin D concentrations, we performed a subgroup analysis using vitamin D cut-offs <30 mmol/L for deficiency, 30–50 mmol/L for insufficiency, and ≥50 mmol/L for sufficiency.

All analyses were performed with IBM SPSS Statistics package (IBM Corp., version 26, Armonk, NY, USA). 

## 3. Results

A total of 912 participants were included in the present analysis. Maternal characteristics are described in Table 1. In summary, average maternal age was 32 years, and average pre-pregnancy BMI was 32.9 kg/m^2^. Of the women, 86% were Caucasian, 25% had a family history of diabetes, and 62% had had previous pregnancies. Two thirds of the participants (66.3%) had total 25OHD concentrations ≥ 50 nmol/L at baseline. VitD concentrations and glucose homeostasis variables according to pregnancy period are summarized in Table 2.

Overall, statistical assumptions for regression analyses were met, with the exception of 3 indices displaying a Durbin–Watson statistic and/or Cook’s distance close to 1 at 24–28 weeks and a single index that showed a Durbin–Watson statistic close to 1 at 35–37 weeks. 

Only the significant associations observed in multivariate analysis between the VitD compounds and the pregnancy-validated glucose homeostasis variables are displayed in Table 3. The associations are schematically summarized in Figure 1. 

At first assessment (<20 weeks’ gestation), 25OHD3 was associated with higher FPG, decreased fasting insulin secretion (HOMA-β) and decreased fasting DI (1/HOMA-IR* HOMA-β and QUICKI* HOMA-β). 

At 24–28 weeks, 25OHD3 was associated with higher FPG and 2 h plasma glucose, lower fasting insulin secretion (HOMA-β), and lower fasting DI (QUCKI* HOMA-β). 25OHD2 was associated with higher post-challenge insulin sensitivity (Matsuda) and higher post-challenge DI (Matsuda*Stumvoll phase 1). 

At 35–37 weeks, 25OHD3 was associated with higher fasting insulin sensitivity (1/HOMA-IR), and 25OHD2 was associated with higher post-challenge DI (Matsuda*Stumvoll phase 1). 

No significant associations between glucose homeostasis variables and C3-epimer were observed at any pregnancy period.

Furthermore, no significant associations between any vitamin D compounds and HiP were observed at any pregnancy period assessed.

Coefficients for the associations were weak or very weak, with global R^2^ changes ranging from 0.010 to 0.144 for the different glucose homeostasis indices. 

No significant associations between the additional glucose homeostasis indices (OGIS, IGI, Stumvoll phase 2, or their calculated disposition indices) were observed. 

In the subgroup analysis, both favorable and unfavorable associations were present in different sufficiency groups, pregnancy periods, and VitD compounds with the exceptions of only favorable associations being observed with 25OHD_2_ or at 35–37 weeks (Appendix A). The associations were not strong.

## 4. Discussion

In this exploratory sub-analysis of the DALI study, we observed direct associations between VitD compounds and insulin sensitivity and inverse associations with fasting insulin secretion throughout pregnancy. The association between VitD and β-cell function (measured as DI) was inverse at fasting and direct post-challenge, while that for glucose concentrations was direct. It is important to note that all coefficients were weak or very weak, suggesting a small contribution from VitD in glucose homeostasis. Most associations were observed with 25OHD3. 

Our observations indicate a very weak but significant association between VitD and fasting (35–37 weeks) and post-challenge (24–28 weeks) insulin sensitivity indices validated in pregnancy. Several observational studies have reported positive associations between VitD concentrations and insulin sensitivity, generally assessed by HOMA-IR and QUICKI [25,53,54]. This association has been confirmed in VitD supplementation trials, both in and outside pregnancy [23,30,47], with insulin sensitivity usually measured in fasted state [25,27]. 

As to insulin secretion, we observed an association between VitD compounds and lower fasting insulin secretion (<20 and 24–28 weeks). While in animal models, the association between VitD deficiency and insulin secretion impairment is well-characterized, human observational and intervention trials tend to focus on insulin sensitivity rather than secretion. Reports involving VitD and insulin secretion have been inconsistent. Small intervention studies have shown increased insulin secretion with supplementation with VitD3 in four subjects with insulin deficiency [55] or with 1-alfa-OH-D3 in seven subjects with diabetes [56]. On the other hand, in line with our observations, significant inverse associations between total 25OHD and OGTT-induced insulin secretion [22] and hyperglycemic clamp-induced insulin response have been reported [24].

As to more relevant β-cell function indices, we observed a consistent inverse association between VitD compounds and fasting DI (QUICKI*HOMA-β in early and mid-pregnancy, 1/HOMA-IR*HOMA-β in early pregnancy) and a direct association with post-challenge DI (Matsuda*Stumvoll phase1) in mid- and late pregnancy. 

A direct association between 25OHD3 and glucose (fasting at <20 and 24–28 weeks, and post-challenge at 24–28 weeks) was observed. 

Though weak, the associations with 25OHD3 can be viewed as unfavorable (mainly reduced fasting insulin, decreased fasting DI, and higher glucose concentrations), while those for 25OHD2 were favorable (increased post-challenge sensitivity (Matsuda) and increased post-challenge β-cell function, estimated by the Matsuda*Stumvoll phase 1 index). 

Reports addressing VitD and glucose homeostasis usually study the associations between 25OHD3 or total 25OHD and different glucose-related variables, but less is known about the influence of individual VitD compounds. An analysis from the European Prospective Investigation into Cancer and Nutrition (EPIC) case–cohort study for type 2 diabetes mellitus involving 9671 incident cases described an inverse association between 25OHD3 and incident type 2 diabetes melitus, suggesting a favorable effect, while C3-epimer concentrations were directly associated with incident type 2 diabetes mellitus. This pointed to a potential negative role of C3-epimer in glucose-homeostasis regulation. There was no statistically significant association with type 2 diabetes mellitus for 25OHD2 in this report [20]. We did not find any significant association between C3-epimer and any glucose homeostasis variable.

It is generally accepted that 25OHD3 has greater biological activity compared with 25OHD2, as it has proven to be more efficient at raising total serum 25OHD concentrations [4]. However, studies comparing health outcomes after different regimes of vitamin supplementation are scarce. Curiously, a study involving mice exclusively fed either VitD2 or VitD3 showed that D2-mice had lower serum 1,25(OH)_2_D relative to D3 mice, but in contrast, free 25OHD was significantly higher. Furthermore, D2-mice had significantly higher bone volume/total volume and trabecular number compared with D3-mice [57]. Thus, the ideal supplementation strategy might not be completely clarified.

Our observations showed a mixed relationship between VitD compounds and glucose homeostasis variables, hinting to a more favorable association with 25OHD2 compared with 25OHD3. However, the magnitudes of the associations observed suggest that VitD does not have a pivotal role in glucose homeostasis regulation in this study group. Nonetheless, results might not be extrapolated to other populations of pregnant women (i.e., non-obese) 

Evidence from recent reports addressing the prevention of type 2 diabetes mellitus have shown a beneficial effect from VitD supplementation [58,59] that seems to be limited to non-obese subjects [59]. More recently, a secondary analysis from the D2d trial (comparing VitD with placebo for the prevention of type 2 diabetes mellitus) investigated the effects of VitD3 supplementation on insulin sensitivity and β-cell function. The authors of the analysis observed that supplementation with VitD3 did not improve insulin sensitivity or β-cell function in people with prediabetes, but there was benefit among those with very low baseline total 25OHD status [60]. In light of these observations, the authors speculated that the potential effect of VitD on the risk of type 2 diabetes mellitus might be mediated through other pathways independent of insulin sensitivity and β-cell function [60]. 

In pregnancy, a Cochrane review examining the effect of VitD supplementation reported a GDM risk reduction (RR 0.51, 95CI 0.27–0.97) [39]. Interestingly, average baseline total 25OHD concentration in the women included in this meta-analysis was <50 nmol/L. We reported similar observations commenting on a metanalysis where VitD supplementation showed a GDM risk reduction. Studies included comprised women with VitD deficiency at baseline and high supplementation doses (~3500 Ui/day) [61]. In this report, we did not observe differences between sufficiency groups that suggest more favorable or unfavorable associations according to total 25OHD concentrations. As abovementioned, this could be due to different relationships between VitD metabolites and glucose homeostasis indices in obese subjects.

We reported different associations between VitD compounds and glucose homeostasis variables. Though our observations hint at a more favorable role for 25OHD2, the magnitudes of the associations observed suggest that vitamin D may not play a key role in glucose homeostasis regulation in pregnancy. Little is known of the associations between individual VitD compounds and glucose metabolism in pregnancy, and further studies are needed in order to define their roles in glucose homeostasis.

### Strengths/Limitations

The main novelty presented in this report is the relationships between different VitD compounds (instead of total VitD) and glucose homeostasis. Another strength is the use of pregnancy-validated sensitivity and secretion indices and the use of the disposition index as a measure of β-cell function, which can be more informative than isolated insulin sensitivity and secretion values.

As for the limitations, this was an exploratory analysis, so the associations observed must be interpreted with care. C3-epimer was not quantitatively measured. Additionally, in our study, only overweight and obese women were included, and the majority of the participants were VitD sufficient at baseline and throughout pregnancy. Considering recent evidence from type 2 diabetes mellitus prevention trials, the characteristics of our population may not have allowed us to fully explore the associations between VitD compounds and glucose homeostasis in pregnancy.

## 5. Conclusions

In women participating in the DALI study, most of them with baseline VitD in sufficiency range, VitD compounds were not unequivocally associated with more favorable glucose homeostasis measures. This suggests a limited potential of VitD compounds for HiP prevention in populations with similar characteristics. Whether different VitD compounds have different impacts on glucose homeostasis is not clear, but further exploring their role might be of interest for the design of supplementation studies.

## Figures and Tables

**Figure 1 nutrients-14-03256-f001:**
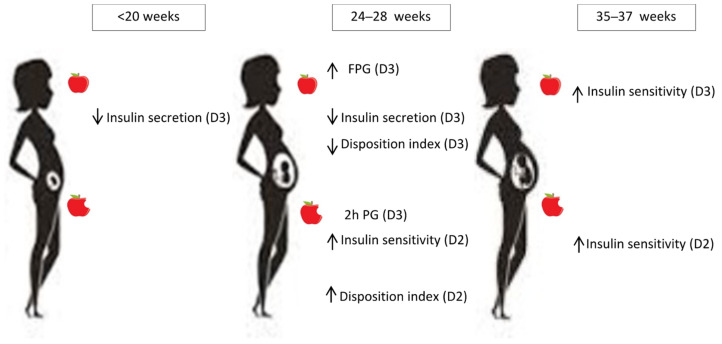
Associations between vitamin D compounds and pregnancy-validated glucose homeostasis variables according to pregnancy period.

**Table 1 nutrients-14-03256-t001:** Maternal characteristics and sociodemographic data.

Variables	Median (P_25–75_) or n (%)	n
Maternal age (years)	32.1 (28.4–36.0)	912
Height (cm)	166 (161–170)	912
Gestational age at baseline (weeks)	15.0 (13.2–16.7)	912
Ethnicity (Caucasian)	784 (86.2)	910
Family history of DM	228 (25.0)	912
Previous pregnancies	567 (62.3)	910
GDM	58 (10.4)	558
Stillbirth	64 (11.5)	558
Congenital anomalies	24 (4.3)	560
Macrosomia	120 (21.6)	556
BMI according to pregnancy period		
Pre-pregnancy BMI (kg/m^2^)	32.9 (30.6–36.2)	912
<20 weeks	33.7 (31.6–36.8)	911
24–28 weeks	34.9 (33.1–37.9)	655
35–37 weeks	36.6 (34.2–39.2)	500

BMI: body mass index, DM: diabetes mellitus, GDM: gestational diabetes mellitus.

**Table 2 nutrients-14-03256-t002:** Vitamin D compounds and pregnancy-validated glucose homeostasis variables according to pregnancy period.

Variable	<20 Weeks		24–28 Weeks		35–37 Weeks	
	Median (P25-75) or n (%)	n	Median (P25-75) or n (%)	n	Median (P25-75) or n (%)	n
**Vitamin D compounds**						
25OHD_total_ (nmol/L)	61.8 (43.5–79.3)	912	70.4 (47.9–98.7)	660	71.1 (45.7–101.4)	502
25OHD_2_ (nmol/L)	0.025 (0.025–8.310) ‡	912	0.025 (0.025–8.299)	660	0.025 (0.025–6.027)	502
25OHD_3_ (nmol/L)	60.1 (42.0–77.2)	912	68.3 (46.8–97.0)	660	67.2 (42.5–99.0)	502
C3-epimer (+)	323 (36%)	896	296 (46%)	643	233 (47.6%)	490
Vitamin D sufficiency (≥50 nmol/L)	605 (66.3%)	912	476 (72.1%)	660	348 (69.3%)	502
**Glucose**						
FPG (mmol/L)	4.7 (4.4–5.0)	912	4.6 (4.3–4.8)	660	4.5 (4.2–4.8)	502
1 h PG (mmol/L)	7.0 (5.8–8.3)	902	7.6 (6.6–8.8)	652	8.2 (7.1–9.2)	485
2 h PG (mmol/L)	6.0 (5.2–6.9)	904	6.2 (5.4–7.0)	652	6.4 (5.6–7.2)	485
**Fasting glucose homeostasis indexes**						
HOMA-IR (mUI/L*mmol/L)	2.8 (2.1–3.9)	892	3.0 (2.2–4.1)	645	3.3 (2.4–4.4)	491
QUICKI (1/(log μUI/mL + log mg/dL)	0.33 (0.31–0.34)	892	0.32 (0.31–0.34)	645	0.32 (0.31–0.33)	491
HOMA-β (mUI/mmol)	237 (177–340)	890	287 (206–405)	642	348 (252–528)	488
Fasting DI						
QUICKI*HOMA-β	76.7 (58.6–108.9)	890	92.9 (67.3–127)	642	111 (81.9–164.5)	488
1/HOMA-IR* HOMA-β	79.8 (60–114)	890	88.9 (72.1–130.8)	642	100 (72.1–153)	488
**Post-challenge glucose homeostasis indexes**						
Matsuda (μUI/mg)	3.1 (2.1–4.3)	408	2.6 (1.9–3.5)	311	2.2 (1.6–2.9)	220
Stumvoll phase 1 (pmol)	1623 (1257–2117)	845	1843 (1368–2347)	622	2200 (1732–2775)	470
AUC_ins/glu_ (μUI/mmol	12.7 (9.0–19.0)	414	14.8 (10.3–19.5)	311	17.6 (13.0–23.4)	224
Post-challenge DI						
Matsuda*AUC_ins/glu_	38.4 (29.8–47.6)	408	34.9 (30.0–43.2)	310	38.1 (29.9–46.1)	220
Matsuda*Stumvoll1	4892 (4029–5979)	408	4595 (3836–5469)	311	4576 (3940–5599)	220
HiP	245 (27.1%)	904	140 (21.5%)	652	97 (20%)	486

‡ Expressed as P_10–90_ due to low concentration, FPG: fasting plasma glucose, PG: plasma glucose, HOMA-IR: homeostasis model assessment insulin resistance, HOMA- β: homeostasis model assessment beta, QUICKI: quantitative insulin sensitivity check index, AUC_ins/glu_: area under the curve insulin/glucose, DI: disposition index, HiP: hyperglycemia in pregnancy, C3-epimer (+): C3-epimer positivity.

**Table 3 nutrients-14-03256-t003:** Associations between vitamin D compounds and pregnancy-validated glucose homeostasis variables in a multivariate model adjusted for age, body mass index, ethnicity, family history of diabetes, prior GDM, recruitment site, and, at 24–28 and 35–37 weeks, DALI lifestyle intervention.

Outcome Variables	β Values/OR for Significant Associations								
	<20 Weeks			24–28 Weeks			35–37 Weeks		
	25OHD2	25OHD3	C3-Epimer	25OHD2	25OHD3	C3-Epimer	25OHD2	25OHD3	C3-Epimer
Fasting									
FPG		0.124 *			0.111 *				
1/HOMA-IR (sens)								0.127 *	
QUICKI (sens)									
HOMA-β (sec)		−0.117 *			−0.145 **				
Fasting DI									
QUICKI* HOMA-β		−0.124 *			−0.148 **				
1/HOMA-IR* HOMA-β		−0.093 *							
Post-challenge									
1 h PG									
2 h PG					0.120*				
Matsuda (sens)				0.149 *					
Stumvoll phase 1 (sec)									
AUC_ins/glu_ (sec)									
Post-challenge DI									
Matsuda*AUC									
Matsuda*Stumvoll1		−0.103 *		0.168 *			0.239 **		
HiP									

* = *p* < 0.05, ** = *p* < 0.01. FPG: fasting plasma glucose, HOMA-IR: homeostasis model assessment insulin resistance, sens: insulin sensitivity index, QUICKI: quantitative insulin sensitivity check index, HOMA- β: homeostasis model assessment beta, sec: insulin secretion index; DI: disposition index, PG: plasma glucose, AUC_ins/glu_: area under the curve insulin/glucose, HiP: Hyperglycemia in pregnancy.

## Data Availability

The data that support the findings of this study are available from the corresponding author, RC, upon reasonable request.

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
