# Peer review of "The Weak Relationship between Vitamin D Compounds and Glucose Homeostasis Measures in Pregnant Women with Obesity: An Exploratory Sub-Analysis of the DALI Study"

_nutrients, 2022, doi:10.3390/nu14163256_

Round 1

Reviewer 1 Report

Manuscript ID: nutrients- 1806379

Authors: LC Mendoza et al.

Title: "Relationship between vitamin D compounds and glucose homeostasis measures in obese pregnant women. An exploratory sub-analysis of the DALI study."

The authors of the present sub-analysis of the DALI study tried to explore the potential associations of VitD compounds with several glucose homeostasis variables in overweight and obese pregnant women. The authors have measured various glucose homeostasis indices at fasting and (OGTT) post-challenge.

According to the study findings, no conclusions could be made regarding VitD compounds and a potentially favorable association with glucose homeostasis.

The following points should be considered.

Major comments:

  1. According to the authors, "Women with normal glucose tolerance were randomized to three lifestyle arms (44), three lifestyle arms vs usual care (45) or lifestyle and/or VitD vs. usual care (38)." Would it be possible for the authors to briefly mention the three lifestyle arms?
  2. Did the authors examine if the study findings are consistent for pregnant women with vitamin D sufficiency, insufficiency, or deficiency?

3.     Moreover, did the authors consider adjusting for family history of diabetes and previous GDM?

  1. Did the authors also consider adjusting for the laboratory (local or central) used to calculate the glucose homeostasis indices?
  2. According to the authors, "Women with <20 weeks' gestation, BMI ≥29 Kg/m2were eligible for the study". Was BMI threshold (≥29) chosen in the original DALI study or for the present exploratory analysis? Please clarify why this cut-off value is applied if the latter is correct.
  3. Did the authors adjust their findings for sex steroid hormones, hsCRP?

Minor comments:

1.      Please check the manuscript for typos or misspellings (e.g., in line 55, I assume the authors mean the word "form" instead of "from", line 106 "Masuda", etc.)

2.     Please clarify if the Vitamin D compounds 25OHD2, 25OHD3 and C3-epimer were centrally measured."

3.     Please briefly mention if any exclusion criteria were applied.

Author Response

Major comments:

  1. According to the authors, "Women with normal glucose tolerance were randomized to three lifestyle arms (44), three lifestyle arms vs usual care (45) or lifestyle and/or VitD vs. usual care (38)." Would it be possible for the authors to briefly mention the three lifestyle arms?

We have currently expanded the description of the three lifestyle arms and VitD interventions, lines 140-144.

  1. Did the authors examine if the study findings are consistent for pregnant women with vitamin D sufficiency, insufficiency, or deficiency?

We thank the reviewer for this advice. We have now run the analysis in the three subgroups of vitamin D sufficiency, insufficiency or deficiency. Results were not definitely concordant but pointing to more favourable associations in the non-sufficient groups. Results are provided in supplementary Table 1, summarized in the results section and discussed under discussion.

  1. Moreover, did the authors consider adjusting for family history of diabetes and previous GDM?

After the reviewers’ suggestion, we have also adjusted for family history of diabetes and pervious GDM.

  1. Did the authors also consider adjusting for the laboratory (local or central) used to calculate the glucose homeostasis indices?

Most glucose values used to calculate glucose homeostasis indices were central. In the occasions where we imputed local glucose values It was not uncommon that a single OGTT glucose value was imputed so that central and local values were combined in a single index. Thus, we have not adjusted for the laboratory.

  1. According to the authors, "Women with <20 weeks' gestation, BMI ≥29 Kg/m2 … were eligible for the study". Was BMI threshold (≥29) chosen in the original DALI study or for the present exploratory analysis? Please clarify why this cut-off value is applied if the latter is correct.

The BMI threshold corresponds to the original DALI study.

  1. Did the authors adjust their findings for sex steroid hormones, hsCRP?

These variables were not measured in the study and we have not adjusted for them.

Minor comments:

  1. Please check the manuscript for typos or misspellings (e.g., in line 55, I assume the authors mean the word "form" instead of "from", line 106 "Masuda", etc.)

We thank the reviewer for the advice, the misspellings are now corrected

  1. Please clarify if the Vitamin D compounds 25OHD2, 25OHD3 and C3-epimer were centrally measured."

We have now specified that vitamin D compounds were centrally measured.

  1. Please briefly mention if any exclusion criteria were applied.

No exclusion criteria, additional to those of the DALI study itself were applied.  

Reviewer 2 Report

This secondary data analysis offers interesting information regarding vitamin D compounds in relationship to glucose homeostasis – e.g., there is NO relationship.  The authors nicely outline the justification for the analyses in the introduction.  However, there are many acronyms which often make the reading tedious.  Some are not defined (for example, DM2 – is this the same as T2DM? Both acronyms are used).  The acronyms for 25-hydroxycholecalciferol (2 and 3) and 1,25 dihydroxycholecalciferol are not defined in the text.  At line 54, hydroxylase should by hydroxylated.  PG is not defined (until after its first use).  I suggest not relying on all of these acronyms and spelling out terms (such as plasma glucose) to facilitate ease of reading. 

At lines 105 and 107, describing the correlations as ‘good’ and ‘best’ does not allow the reader to assess the strength of the relationship.  Please use the established categories weak-moderate-strong.  At line 108, the citation should state, Powe et al. 

Please state that those signing a consent but who were excluded from the original trial allowed for their data to be used in secondary analyses. 

Please provide more detail on the ‘disposition indices’ (line 167).  It appears based on table 1 that there are 4 possible measures for DI – but it is not always clear what measure is being discussed when the term DI is used.  See line 37 and 269 for example.   Please state the specific indices and not the general term, DI (this was nicely stated in ‘results’ but not in the abstract and discussion). 

It is not clear what the asterisk or double cross refer to in Table 2.  In Table 3 it is not clear what the backward paragraph symbol stands for. 

Were there any outliers in the data sets?  (e.g., were the statistical assumptions followed for the regression analyses?)

In the discussion the authors importantly state that all coefficients are ‘weak’ – actually, they are well below ‘weak’ (e.g., < 0.3) and basically nonexistent.  Hence the statement (line 252) that ‘our observations indicate an association between VitD and fasting and post-challenge insulin sensitivity indices in pregnancy’ is overstating the results.  As is the statement that ‘our observations showed a complex relationship between VitD compounds and glucose homeostasis variables, confirming improved sensitivity….’ (line 299).  The authors need to emphasize throughout the discussion that there does not seem to be a role for VitD in glucose homeostasis.   (This is stated at line 303 and the last sentence of the abstract – hence the text is somewhat contradictively written.)  The title of the paper should emphasize a lack of a relationship as well. 

Author Response

This secondary data analysis offers interesting information regarding vitamin D compounds in relationship to glucose homeostasis – e.g., there is NO relationship.  The authors nicely outline the justification for the analyses in the introduction.  However, there are many acronyms which often make the reading tedious.  Some are not defined (for example, DM2 – is this the same as T2DM? Both acronyms are used).  The acronyms for 25-hydroxycholecalciferol (2 and 3) and 1,25 dihydroxycholecalciferol are not defined in the text.  At line 54, hydroxylase should by hydroxylated.  PG is not defined (until after its first use).  I suggest not relying on all of these acronyms and spelling out terms (such as plasma glucose) to facilitate ease of reading. 

We thank the reviewer for the advice. Hydroxylase is now corrected to hydroxylated, and 25OHD described at first use. We have used type 2 diabetes and plasma glucose instead of the corresponding acronyms in the main text. As to 25-hydroxycholecalciferol and 1,25 dihydroxycholecalciferol, we have defined the acronyms at first use,

At lines 105 and 107, describing the correlations as ‘good’ and ‘best’ does not allow the reader to assess the strength of the relationship.  Please use the established categories weak-moderate-strong.  At line 108, the citation should state, Powe et al. 

            Modifications have been introduced.

Please state that those signing a consent but who were excluded from the original trial allowed for their data to be used in secondary analyses. 

This is now stated.

Please provide more detail on the ‘disposition indices’ (line 167).  It appears based on table 1 that there are 4 possible measures for DI – but it is not always clear what measure is being discussed when the term DI is used.  See line 37 and 269 for example.   Please state the specific indices and not the general term, DI (this was nicely stated in ‘results’ but not in the abstract and discussion). 

In the methods section we have now described all analysed disposition indices and in the abstract and discussion, we have indicated the specific indices that were discussed.

It is not clear what the asterisk or double cross refer to in Table 2.  In Table 3 it is not clear what the backward paragraph symbol stands for. 

We have now indicated the meaning of the signs used in Table 2 and Table 3.

Were there any outliers in the data sets?  (e.g., were the statistical assumptions followed for the regression analyses?)

We have now indicated in the methods section that that statistical assumptions for linear regressions were checked and corresponding results in the results section

In the discussion the authors importantly state that all coefficients are ‘weak’ – actually, they are well below ‘weak’ (e.g., < 0.3) and basically nonexistent.  Hence the statement (line 252) that ‘our observations indicate an association between VitD and fasting and post-challenge insulin sensitivity indices in pregnancy’ is overstating the results.  As is the statement that ‘our observations showed a complex relationship between VitD compounds and glucose homeostasis variables, confirming improved sensitivity….’ (line 299).  The authors need to emphasize throughout the discussion that there does not seem to be a role for VitD in glucose homeostasis.   (This is stated at line 303 and the last sentence of the abstract – hence the text is somewhat contradictively written.)  The title of the paper should emphasize a lack of a relationship as well. 

We have modified the title and modified the text according to the indications of the reviewer.

Round 2

Reviewer 1 Report

Manuscript ID: nutrients- 1806379

Authors: LC Mendoza et al.

Title: "Weak relationship between vitamin D compounds and glucose homeostasis measures in pregnant women with obesity. An exploratory sub-analysis of the DALI study"

The authors have satisfactorily responded to my comments and suggestions. They have made the necessary changes to the manuscript and improved the overall quality of their paper. A few minor comments:

1.     Given the significant hormonal changes during pregnancy and the potential association of sex steroid hormones with glucose homeostasis, I would suggest including the lack of measurement of these variables as a potential limitation of this study in the relevant limitation paragraph.

2.     Please add the reference used for classifying vitamin D levels (deficiency, insufficiency, sufficiency)